# Can Severe Plastic Deformation Tune Nanocrystallization in Fe-Based Metallic Glasses?

**DOI:** 10.3390/ma16031260

**Published:** 2023-02-01

**Authors:** Monika Antoni, Florian Spieckermann, Niklas Plutta, Christoph Gammer, Marlene Kapp, Parthiban Ramasamy, Christian Polak, Reinhard Pippan, Michael J. Zehetbauer, Jürgen Eckert

**Affiliations:** 1Erich Schmid Institute of Materials Science, Austrian Academy of Sciences, 8700 Leoben, Austria; 2Department of Materials Science, Chair of Materials Physics, Montanuniversität Leoben, 8700 Leoben, Austria; 3Department of Mechanical Engineering, University of Dunaujvaros, 2400 Dunaujvaros, Hungary; 4Vacuumschmelze GmbH & Co. KG (VAC) Rapid Solidification Technology, 63450 Hanau, Germany; 5Research Group Physics of Nanostructured Materials, Faculty of Physics, University of Vienna, 1090 Wien, Austria

**Keywords:** severe plastic deformation, amorphous alloys, nanocrystallization

## Abstract

The effects of severe plastic deformation (SPD) by means of high-pressure torsion (HPT) on the structural properties of the two iron-based metallic glasses Fe_73.9_Cu_1_Nb_3_Si_15.5_B_6.6_ and Fe_81.2_Co_4_Si_0.5_B_9.5_P_4_Cu_0.8_ have been investigated and compared. While for Fe_73.9_Cu_1_Nb_3_Si_15.5_B_6.6_, HPT processing allows us to extend the known consolidation and deformation ranges, HPT processing of Fe_81.2_Co_4_Si_0.5_B_9.5_P_4_Cu_0.8_ for the first time ever achieves consolidation and deformation with a minimum number of cracks. Using numerous analyses such as X-ray diffraction, dynamic mechanical analyses, and differential scanning calorimetry, as well as optical and transmission electron microscopy, clearly reveals that Fe_81.2_Co_4_Si_0.5_B_9.5_P_4_Cu_0.8_ exhibits HPT-induced crystallization phenomena, while Fe_73.9_Cu_1_Nb_3_Si_15.5_B_6.6_ does not crystallize even at the highest HPT-deformation degrees applied. The reasons for these findings are discussed in terms of differences in the deformation energies expended, and the number and composition of the individual crystalline phases formed. The results appear promising for obtaining improved magnetic properties of glassy alloys without additional thermal treatment.

## 1. Introduction

In recent times, it has become increasingly important to focus on ways to consume electric energy more efficiently. As magnetic losses—proportional to the area of the B-H-loop—play a significant role in energy consumption (e.g., [1]), it is vital to gain a better understanding of how they can be controlled. Already 40 years ago, it has been found that Fe-based metallic glasses are highly efficient soft magnetic alloys (e.g., [2,3]), especially with respect to achieving low coercivity. Since then, further efforts have been undertaken to improve the magnetic and mechanical properties of such alloys and to establish synthesis routes allowing for larger materials dimensions (e.g., [4,5,6,7,8,9]). The latter is most important, as upscaling melt-spun tapes (thickness of only ~20 µm) would extend their applicability to, e.g., electric engines, and could thereby contribute to a greener future by globally reducing the CO_2_ footprint of electrical devices.

To tune the magnetic properties of these alloys, annealing in an external magnetic field and under external stress is applied (e.g., [6,10,11,12]). Thereby, crystallization of α-Fe und Fe_3_Si [13] from the amorphous phase is induced, which increases the otherwise rather limited saturation polarization (e.g., [3]).

Following the group of Aronin [14], plastic deformation—at best the various methods of SPD (Severe Plastic Deformation) because of the suppression of cracks by the enhanced hydrostatic pressure component (e.g., [15,16])—can provide sufficient structural inhomogeneities as nuclei for controlled nanocrystallization due to a competition of homogeneous deformation and shear banding [17]. Recent results indicate that the hydrostatic pressure during HPT (e.g., [15,18]) may induce local ordering during the deformation process [19] which could further ease the nucleation in some alloy systems.

The most prominent candidate among soft magnetic Fe-based metallic glasses is Finemet/Vitroperm (tradenames Hitachi and VAC, respectively). This material with composition Fe_73.5_Cu_1_Nb_3_Si_13.5_B_9_ is available as thin amorphous ribbons which become nanocrystalline (although brittle) after heat treatment (at approximately 510 °C (783 K)). In this alloy, the elements Cu and Nb play a major role in the nanocrystallization process [20].

If deformation-induced nanocrystallization could be achieved, one could resign on Cu and Nb additions, rendering a higher amount of Fe for providing a high magnetization while nanocrystals—with sizes smaller than 50 nm—still achieve a low coercivity [21]. Indeed, the Aronin group [14] reached a high saturation polarization although the coercivity was considerable. The latter may be reduced by adequate heat treatment [22]. As SPD has some potential for amorphization, also the non-magnetic additives Si and B could be omitted, again increasing the percentage of Fe and thus increasing the low saturation polarization of 1.4 T, thereby making the material more attractive for commercial applications. Makino et al. [23] showed that additions of P and Co—thus providing alloys like Fe_83.3–84.3_Si_4_B_8_P_3–4_Cu_0.7_ and Fe_81.2_Co_4_Si_0.5_B_9.5_P_4_Cu_0.8_, achieve a saturation polarization of 1.88–1.94 T, but so far these alloys were not produced with the appropriate nanosized microstructure. Therefore, applying SPD processing appears as a very promising route to reach nanocrystallinity, providing low coercivity and high saturation polarization at the same time.

Because of the very high hardness of the Vitroperm alloy Fe_73.9_Cu_1_Nb_3_Si_15.5_B_6.6_ (6–10 GPa) and particularly of Makino’s alloy Fe_81.2_Co_4_Si_0.5_B_9.5_P_4_Cu_0.8_ (up to 16 GPa at least when fully crystallized), the feature of SPD methods utilizing high hydrostatic pressures up to about 10 GPa seems essential to provide any plastic deformation in these alloys (e.g., [5,24,25,26]). So far, some amount of deformation has been achieved by ball milling [27], impact hammering [28]) and HPT (e.g., [14,29,30,31]) but these efforts were so far limited to very small samples. However, SPD methods are basically bottom-up methods to achieve *bulk* nanocrystalline and/or amorphous metals and alloys, they are not only capable of nanocrystallization but can even realize massive materials from amorphous ribbons (e.g., [15]).

In a recent work [32] we proved this for the example of Vitroperm Fe_73.9_Cu_1_Nb_3_Si_15.5_B_6.6_ applying HPT with pressures of 7.5 GPa using deformation temperatures *T_de_*_f_ = 473, 573, and 673 K for one turn. In the present paper, we report on further investigations of HPT effects applied on this glass but also present results on the ‘Makino’-type alloy Fe_81.2_Co_4_Si_0.5_B_9.5_P_4_Cu_0.8_ because of the motivation given above. Of special interest is the question of whether the application of HPT provides some nanocrystallization thus allowing for the replacement of Cu and Nb and/or for some further thermal treatment, to finally achieve an excellent soft magnetic material with maximum saturation polarization and minimum coercivity.

## 2. Materials and Methods

Amorphous Fe_73.9_Cu_1_Nb_3_Si_15.5_B_6.6_ and Fe_81.2_Co_4_Si_0.5_B_9.5_P_4_Cu_0.8_ ribbons (called thereafter alloy A, and alloy B, respectively) fabricated by single-roller melt spinning were supplied by Vacuumschmelze GmbH, Hanau, Germany.

First, thermomechanical analyses were performed on the amorphous tape material using differential scanning calorimetry (DSC) and dynamic mechanical analysis (DMA). The characteristic temperatures for the crystallization *T_x_*_,*n*_ as well as their respective activation enthalpies *E_a_*_,*n*_ were determined by DSC (Pegasus 404 F1, Netzsch, Selb, Germany). For these investigations, heating rates of 5, 10, 20, and 50 K/min were used in a temperature range of 323–1273 K. The background correction of the DSC measurements was carried out by subtracting two consecutive heating cycles and using the second cycle as a baseline.

Second, dynamic mechanical analysis (DMA), applied by the Discovery Hybrid Rheometer DHR 3 (TA Instruments) in tension mode, was used to determine the glass transition temperature *T_g_*, which is difficult to be exactly determined by DSC because, for most Fe-based amorphous alloys, for the change in heat capacity during the glass, the transition is small (e.g., [24]). The measurements were conducted for different frequencies in a range of 0.1–10 Hz in a temperature range of 313–873 K.

For the HPT experiments, a custom-made facility with 400 kN capacity, equipped with an induction heating system (e.g., [33,34]) was used (Figure 1). 22 µm thick ribbons were cut in squares of 12 × 12 mm^2^ from both alloys and stacked between two tungsten carbide HPT anvils with a flat top of 8 mm diameter.

The samples were deformed in torsion at the conditions according to Table 1. The deformation temperature, *T_def_*, the maximum number of turns, *N*, and the applicable pressure, *p*, were limited by the material properties of the tungsten carbide anvils.

Macro torsional shear strains, *γ_torsion_*, were calculated according to Equation (1) (see [34,35]):(1)$$\gamma_{torsion}=\;\frac{2\pi Nr}{t}$$

The thickness of the HPT disc, *t*, from which the samples for the consecutive measurements were taken, was evaluated by light microscopy. The torsional shear strains and corresponding disc radii, *r*, as well as the number of HPT turns, *N,* are listed in Appendix A of this paper (see Table A1 for alloy A and Table A2 for alloy B).

X-ray diffraction (XRD) measurements were performed using a benchtop XRD system with Co K_α_ anode and Fe K_β_-filter (Bruker D2 Phaser, Karlsruhe, Germany). After the experiment, the background and K_α2_ contribution were separated from the measured data. The crystallite size *D* was analyzed via the Scherrer Equation (2) where K is the shape factor, λ is the X-ray wavelength, β is the full width at the half maximum of the investigated peak and θ is the position of that peak (see e.g., [36] for more details).
(2)$$D=\frac{K*\lambda }{\beta *\cos\left(\theta \right)}$$

Cross-sections of the HPT-processed samples were mechanically polished for analysis by optical light microscopy (Olympus BX 51 M) and scanning electron microscopy (LEO 1525, Zeiss, Oberkochen, Germany) to investigate the homogeneity of deformation after HPT and possible crack formation. Investigation of ultrafine-grained/nanocrystalline microstructures required careful transmission electron microscopy (TEM) analyses. For this purpose, FIB lamellae were prepared using an FEI Helios NanoLab 660 dual-beam workstation. The microscope was run at voltages 30–2 kV and Ga+ currents ranging from 50 nA (coarse milling) to 50 pA (polishing). High-resolution TEM images, selected area diffraction patterns, and bright-field scanning TEM (STEM) micrographs were recorded using an FEI Themis TEM. The microscope was operated at an accelerating voltage of 300 kV and a probe current of 0.1 nA. Lamellae were cut 1 mm from the samples’ edges.

## 3. Results

### 3.1. Alloy A—Fe_73.9_Cu_1_Nb_3_Si_15.5_B_6.6_

In an earlier work [32] we investigated the effect of HPT deformation on the Fe_73.9_Cu_1_Nb_3_Si_15.5_B_6.6_ alloy. Because of the high strength of the ribbons [25], HPT deformation was limited to one turn, irrespective of pressure and temperature. In the present work, we present an optimized setup, using flat tungsten carbide anvils, which allows us to observe the microstructural changes up to much larger torsional shear strains.

After HPT deformation, the samples were checked for homogeneity by analyzing the cross-section by optical microscopy (Figure 2).

The elliptic shape of the HPT sample is a consequence of the flow pattern, when flat HPT anvils are used for the deformation procedure. However, as visible from the light microscope image even for the highest degree of deformation (e.g., *N* = 10 and *γ_torsion_* = ~3400), no large-scale cracking did occur, probably due to the fact that the tungsten carbide anvils allow us to provide a maximum HPT pressure of 4 GPa. Moreover, it seems that the increase of the HPT deformation temperature to 473 K (about 60% of the glass transition temperature) provides sufficient deformability, thus limiting the crack length to a few submicrons in size (compare [32]). 60% of the glass transition temperature corresponds to the region where metallic glasses typically show a transition from inhomogeneous to more homogeneous deformation [37].

At first, undeformed samples were investigated by XRD and compared to the deformed material after various HPT turns. The diffraction pattern of the undeformed state in Figure 3 does not change with an increasing number of HPT turns (*N* = 2, 5, 10), suggesting that no crystallization occurred. Thus, even though the samples have been HPT-deformed at 473 K (see Table 1) alloy A still remains amorphous.

Earlier DMA experiments on the undeformed state (as-spun Fe_73.9_Cu_1_Nb_3_Si_15.5_B_6.6_ ribbons) [32] revealed a glass transition temperature *T_g_* of 740 ± 0.5 K [heating rate of 10 K/min] with the corresponding activation enthalpy *E_a_* being 282 ± 28 kJ/mol.

The crystallization kinetics was further studied by DSC measurements in the temperature range from 323 K to 1273 K using a heating rate of 20 K/min. The DSC scans in Figure 4 compare HPT-deformed samples at *N* = 2, 5, 10 turns with the as-spun ribbon material. The two peak temperatures, *T_x_*_1_ = 808 ± 5 K and *T_x_*_2_ = 983 ± 5 K were found to have about the same activation enthalpy of 470 ± 20 kJ/mol (see [32]). According to literature (e.g., [13,21,38]) the first stage of crystallization corresponds to the precipitation of α-Fe and the intermetallic phase Fe_3_Si whereas the second one is related to the crystallization of the residual glass and the formation of Fe-borides. The temperatures of the peaks do not shift with a changing number of HPT turns, indicating that the microstructure beyond a certain degree of HPT deformation does not change anymore. This result agrees well with the XRD results (see Figure 3) where even the sample deformed at the highest number of turns (*N* = 10, *γ_torsion_* = ~3400) still shows an amorphous profile similar to that of the undeformed as-quenched state.

Further evidence for the amorphous state of alloy A after HPT deformation is provided by high-resolution TEM analyses. Figure 5a shows a high-resolution TEM image of a FIB lamella prepared from the highest deformation state (*N* = 10 corresponding *γ_torsion_* = ~3400). The uniform contrast therein clearly shows that no crystals formed at this deformation state. Further, the selected area diffraction pattern in Figure 5b appears blurred, indicating that the alloy stayed amorphous. In sum, the DSC measurements, as well as the XRD and TEM analyses confirm that alloy A exhibits no crystallization upon HPT deformation.

### 3.2. Alloy B—Fe_81.2_Co_4_Si_0.5_B_9.5_P_4_Cu_0.8_

Studying the phase evolution of Fe_81.2_Co_4_Si_0.5_B_9.5_P_4_Cu_0.8_ after the same number of HPT turns as imposed to alloy A but at 293 K reveals a completely different behavior. Figure 6 reveals that crystalline α-Fe ((110), (200), (211), (220)) develops with increasing HPT strain. The intensity of the crystalline peaks steadily increases after the first appearance after 2 HPT turns. These results suggest that with α-Fe only a single phase crystallizes. It is the desired soft magnetic phase; no peaks corresponding to the hard magnetic phases (e.g., borides) are visible. Furthermore, by means of the Scherrer Equation (2), a grain size of about 9 nm was calculated for the sample deformed to 10 turns, thus to the highest imposed strain.

The thermal characteristics of alloy B before HPT deformation were investigated by DSC and DMA. DSC was carried out by heating from room temperature to 1273 K at heating rates between 5 and 50 K/min, to evaluate the temperatures *T_x,1_* and *T_x,2_* at the peak positions of the first and second stage of crystallization, respectively. Both peak temperatures rise with increasing heating rate (Figure 7a) which—by means of Kissinger analysis [39]—allowed to determine the activation enthalpies as *E_a_*_1_ = 240 ± 13 kJ/mol and *E_a_*_2_ = 897 ± 263 kJ/mol.

An in-depth analysis of the glass transition behavior of the undeformed state of alloy B was conducted by DMA measurements, to the authors’ best knowledge for the first time. The storage modulus was measured at frequencies ranging from 0.1–10 Hz at a heating rate of 10 K/min (Figure 8a). The glass transition temperature *T_g_* of alloy B (Fe_81.2_Co_4_Si_0.5_B_9.5_P_4_Cu_0.8_) was determined from the onset of the storage modulus drop (see arrows in Figure 8a). It rises with increasing frequency, yielding a glass transition temperature of *T_g_* = 526 ± 0.5 K measured at 0.1 Hz. The corresponding activation enthalpy was determined as *E_a,g_* = 151 ± 3 kJ/mol using the slope of the Arrhenius plot (Figure 8b). The strong linearity in the Arrhenius plot indicates that alloy B is a strong glass former in contrast to a fragile glass former that would show Vogel-Fulcher-Tammann-like behavior [40].

To study the crystallization behavior of the HPT-deformed samples of alloy B, DSC measurements were conducted after a different number of HPT turns (*N* = 1, 2, 5, and 10) and compared with the undeformed (as-quenched) state. HPT deformation was conducted at room temperature and under a pressure of 3 gPa according to Table 1.

In contrast to alloy A, alloy B shows a limited deformation behavior and higher brittleness, probably because of the deformation-induced crystallinity during HPT (compare remarks on hardness in Section 1, and refs. [5,25,26]). Light microscopy of the sample cross-sections revealed that the material is not crack-free (Figure 9). The crack in the center of the HPT disc spans over the entire sample cross-section and developed during the unloading of the HPT anvils.

The DSC curves in Figure 10 were collected at a heating rate of 20 K/min starting at room temperature up to 1273 K. With an increasing number of turns, the first peak shifts to lower temperatures, from 675 ± 5 K in the undeformed state to 668 ± 5 K after 5 turns, but the crystallization enthalpy decreases with increasing strain, entirely disappearing after 10 turns. This means that crystallization of the phase associated with that peak already occurs during HPT deformation, with a maximum of HPT-induced crystallization after 10 turns. The crystallization enthalpy of the second DSC peak does not change significantly with an increasing number of HPT turns, indicating that no HPT-induced crystallization of additional phases occurs.

For a direct structural analysis, STEM images were taken from a FIB lamella prepared from the highest deformation state achieved at (*N* = 10, *γ_torsion_* = ~2500). Figure 11a unambiguously shows that numerous crystals with a size of 6 (±1) nm develop in alloy B, covering about 50% of the whole sample volume. This crystal size matches the size estimated by the Scherrer equation (see Section 2). Figure 11b presents the corresponding selected-area diffraction pattern. There is an increased number of rings with slightly varying intensity, indicating nanocrystallization with minor orientation anisotropy.

## 4. Discussion

Summing up the results demonstrated in detail in the foregoing section clearly shows that the elastic and plastic deformation under elevated hydrostatic pressure—as it is usually provided by HPT—induces crystallization in alloy B (Fe_81.2_Co_4_Si_0.5_B_9.5_P_4_Cu_0.8_). Alloy A (Fe_73.9_Cu_1_Nb_3_Si_15.5_B_6.6_) under similar conditions, however, stays amorphous at comparable as well as very high shear strains. In what follows, the reason for this very different behavior of alloys A and B will be discussed.

First, we wish to emphasize that—for the example of alloy B—SPD driven crystallization has been convincingly demonstrated for the first time, by complementary evidence from differential scanning calorimetry, dynamic mechanical analysis, X-ray diffraction, and transmission electron microscopy. With the help of the latter results, one can even quantify the crystallization effect induced by HPT by estimating the volume fraction of crystalline phase as a function of applied HPT shear strain using the measured DSC and XRD data shown in Figure 7 and Figure 8, respectively.

The XRD measurements (Figure 6) of alloy B show that the undeformed state is completely amorphous. Kuhnt [5] reported that the maximum achievable crystalline volume fraction in alloy B does not exceed 55 vol.%. Therefore, we assumed a maximum crystallization of 55 vol.% for the sample after 10 HPT turns, because at this stage of HPT deformation the first DSC peak in Figure 7 corresponding to precipitation of the soft magnetic α-Fe phase (e.g., [13]) vanishes completely. By scaling the crystallization enthalpy—i.e., the area of this peak for *N* = 1, 2, 5, and 10 turns—with that of the undeformed state (100% amorphous = maximum area), the evolution of the crystallized volume fraction as a function of HPT deformation can be estimated, results of which are given in Figure 12.

Now, to understand the different effects of HPT on alloys A and B in question, let us consider the energy which is expended mechanically during HPT (*E_exp_*) in each of both cases. For this purpose, in-situ HPT torque measurements of Vorhauer and Pippan [41] and Schniewind [42] can be used, which both show saturation of the torsional shear stress at least at the torsional shear strains achieved in this work. The latter can be determined from the sample thicknesses as well as the number of HPT turns, and then the *E_ex_*_p_ values can be calculated from the fairly rectangular areas below the stress-strain curves (for details, see Appendix A). The resulting values of *E_exp_* are shown as a function of strain for both alloys in Figure 13. There it can be seen that in alloy B (i) *E_exp_* is larger at all torsional shear strains applied, and also that (ii) the alloy-specific activation E_a_ = 472 ± 10 kJ/mol is reached already at *γ_torsion_* = 36. These two facts may explain why in this alloy crystallization occurs easier than in alloy A where the values of expended energy at all torsional shear strains applied are smaller. This is not an effect of apparently different deformation temperatures which—in terms of their specific glass temperatures—are quite similar (alloy A: *T_H_* = 0.57, alloy B: *T_H_* = 0.64)—with T_H_ here being the deformation Temperature T_def_ given in units of the glass transition temperature T_g_. More significantly seems to be that the activation enthalpy for crystallization in alloy A is almost twice as high—E_a_ = 472 ± 10 kJ/mol—than that of alloy B being *E_a_* = 240 ± 13 kJ/mol. However, getting into detail with this explanation, two problems appear: (i) the crystallization in alloy B only appears beyond an HPT strain of *γ_torsion_* = ~580 and not already at *γ_torsion_* = 36, and (ii) deformation-induced crystallization should occur also in alloy A: the critical activation enthalpy of E_a_ = 472 ± 10 kJ/mol should be reached here at torsional strains of about *γ_torsion_* = ~90. However, even at the highest *γ_torsion_* = 3400 applied, crystallization did not occur (unless both the applied pressure and deformation temperature are significantly higher, for details see [32]). At this point we favor another explanation: Alloy A crystallizes in several phases, mainly in two ones, namely α-Fe and the intermetallic phase Fe_3_Si with large differences in the phase compositions requiring complex nucleation processes which cannot be assisted by plastic deformation alone, and hence represent a kinetic obstacle for the crystallization process. In contrast to that, in the case of alloy B, only one crystallization peak of the α-Fe phase is found (e.g., see [4,26]), which has a comparably small difference to the initial phase composition and which can be easily overcome by deformation energy in the kinetic window provided by the HPT process. This explanation is confirmed if we inspect the existing literature on SPD-induced nanocrystallization: Several melt-spun Al alloys [43,44] which show only one crystallization peak during heating, exhibit crystallization during/after HPT processing; the same is true for a CuZrTi glass [45]. In contrast to that, all the investigations on Vitroperm-type alloys show SPD-induced crystallization only at higher pressures (at least 6 gPa [46]) or higher pressures (7.5 gPa) and elevated temperatures equal to or larger than 300 °C (573 K) [32]. There seems to be only one exception represented by the work of Aronin et al. [14,29] who reported substantial deformation-induced nanocrystallization at RT in Fe_78_Si_13_B_9_ which however—in contrast to alloy A—does not contain Cu and Nb. This alloy crystallizes in two stages with activation energies of E_a_ ~325 ± 5 K for both [47]. Compared to alloy A, this indicates an easier start of the crystallization process, and compared to alloy B a harder start of the crystallization process.

Finally, it should be emphasized that these features should allow for significant improvements in the magnetic properties of Fe-based glasses, at least of the ‘Makino’-type-alloy Fe_81.2_Co_4_Si_0.5_B_9.5_P_4_Cu_0.8_. Related results will be reported in a forthcoming paper [48].

## 5. Summary and Conclusions

Concerning energy efficiency for transformer and sensor applications, iron-based metallic glasses are among the best soft magnetic materials because of their low coercivity. However, saturation polarization is known to be limited. Former strategies to specifically heat the material for partial nanocrystallization were successful; however, the whole preparation process appears extensive and rather complicated for commercial production. Therefore, this paper considered the potential of SPD to trigger crystallization, thus simplifying the processing scheme. For this purpose, this paper investigated the microstructural response of two important glassy alloys, A (Fe_73.9_Cu_1_Nb_3_Si_15.5_B_6.6_ “Vitroperm”) and B (Fe_81.2_Co_4_Si_0.5_B_9.5_P_4_Cu_0.8_ “Makino”). Efforts at first focused on the realization of bulk glassy samples by HPT, by consolidation from amorphous melt-spun ribbons and/or quenched foils. In a previous paper, we have shown that applying HPT to alloy A (Fe_73.9_Cu_1_Nb_3_Si_15.5_B_6_) fulfills this goal. With this paper, we showed that the thickness of the sample can be even further increased by extension of the HPT parameters and that also other glassy alloys (alloy B, “Makino”) can be turned into massive shapes. Further results are as follows:For alloy B (“Makino” Fe_81.2_Co_4_Si_0.5_B_9.5_P_4_Cu_0.8_) it is even possible to partially crystallize it as α-Fe precipitates at room temperature by means of HPT, with an upper limit of about 50 vol.% crystallinity of crystals with about 6 nm size. This crystal size is significantly lower than that received so far by conventional annealing procedures; moreover, HPT opens better possibilities for precise tuning of crystallinity and crystal size, which seems important for the tuning of the magnetic properties as well.For alloy A (“Vitroperm” Fe_73.9_Cu_1_Nb_3_Si_15.5_B_6.6_), HPT crystallization is not possible even at the highest HPT parameter values chosen, because of the quasi-simultaneous crystallization of two phases and compositional differences between crystalline and amorphous phases in case of alloy A (“Vitroperm”) compared to the case of alloy B (“Makino”) alloy. These large differences result in a kinetic barrier that cannot be overcome by HPT deformation.

As already mentioned, a forthcoming paper will report on the HPT-induced changes in magnetic properties [48].

## Figures and Tables

**Figure 1 materials-16-01260-f001:**
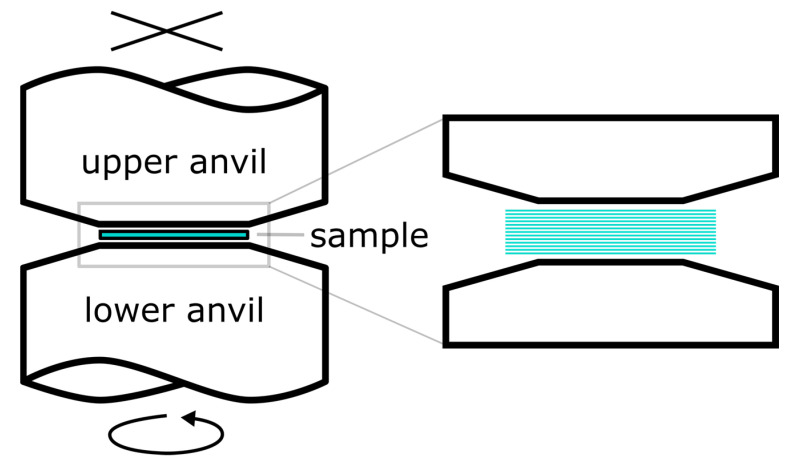
Working principle of HPT with flat top anvils, up to 50 ribbons were stacked between the anvils for HPT consolidation.

**Figure 2 materials-16-01260-f002:**
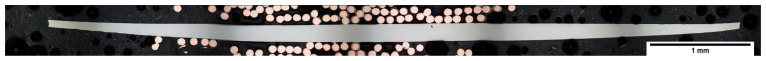
Cross-section of Fe_73.9_Cu_1_Nb_3_Si_15.5_B_6.6_ after HPT-deformation at 473 K and 10 turns.

**Figure 3 materials-16-01260-f003:**
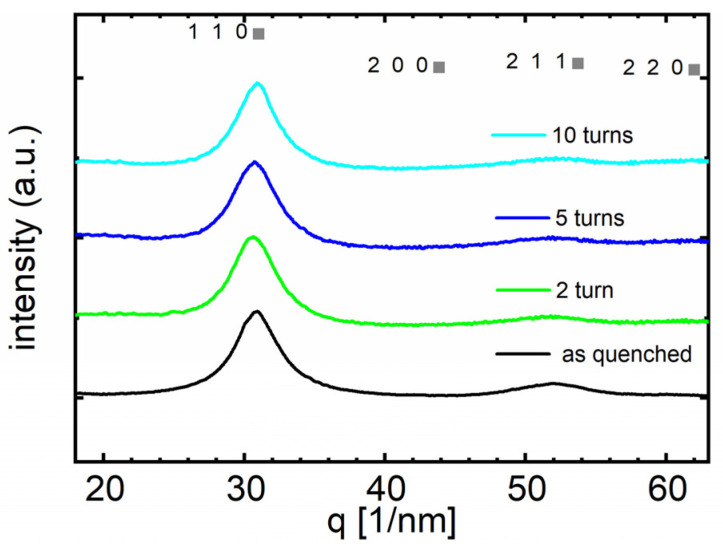
X-ray diffraction patterns of alloy A (Fe_73.9_Cu_1_Nb_3_Si_15.5_B_6.6_) before (as-quenched) and after HPT-deformation at *N* = 2, 5, and 10. The positions of the expected crystalline peaks for α-Fe are indicated to highlight that the sample remains amorphous.

**Figure 4 materials-16-01260-f004:**
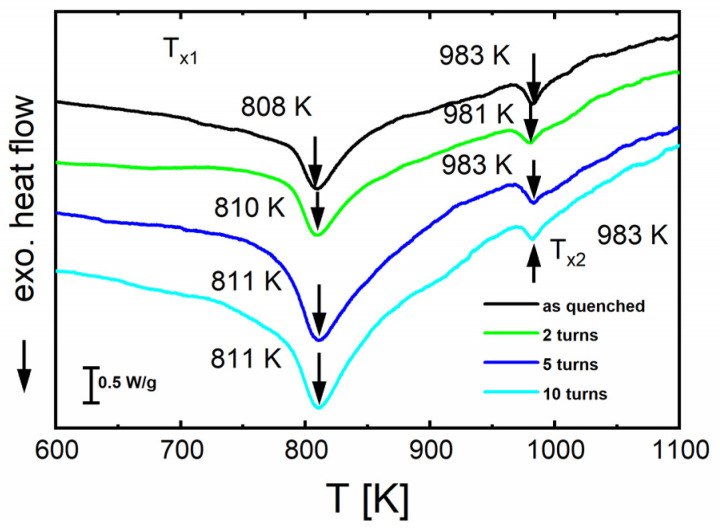
Constant-rate heating DSC scans (heating rate 20 K/min) of alloy A (Fe_73.9_Cu_1_Nb_3_Si_15.5_B_6.6_) before (as-quenched) and after HPT deformation at *N* = 2, 5, and 10.

**Figure 5 materials-16-01260-f005:**
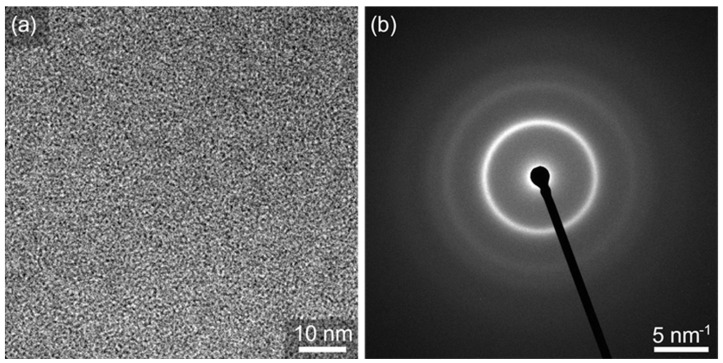
High-resolution TEM image of a FIB-lamellae (**a**) and the corresponding selected-area diffraction pattern (**b**) of alloy A (Fe_73.9_Cu_1_Nb_3_Si_15.5_B_6.6_) deformed by HPT at 473 K for 10 turns.

**Figure 6 materials-16-01260-f006:**
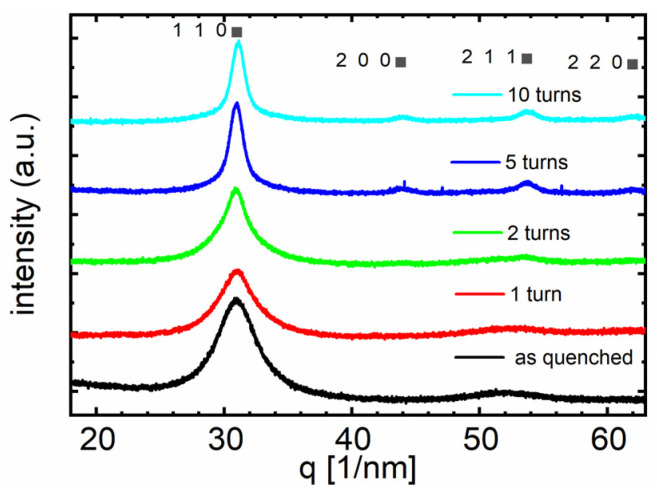
X-ray diffraction patterns of alloy B (Fe_81.2_Co_4_Si_0.5_B_9.5_P_4_Cu_0.8_) before and after HPT deformation (*N* = 1, 2, 5, and 10) at room temperature.

**Figure 7 materials-16-01260-f007:**
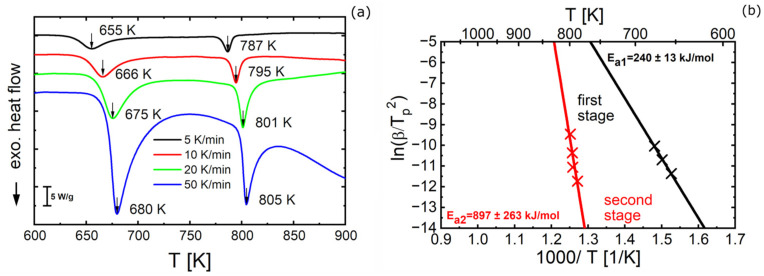
DSC measurements for alloy B (Fe_81.2_Co_4_Si_0.5_B_9.5_P_4_Cu_0.8_) before HPT deformation (**a**) DSC scans obtained at different heating rates (5, 10, 20, and 50 K/min) and (**b**) Kissinger plots for determination of the activation enthalpies *E_(ai)_* for both crystallization stages. Β denotes the heating rate in K/min and *T_p_* the local minimum of the respective stage of crystallization for that heating rate.

**Figure 8 materials-16-01260-f008:**
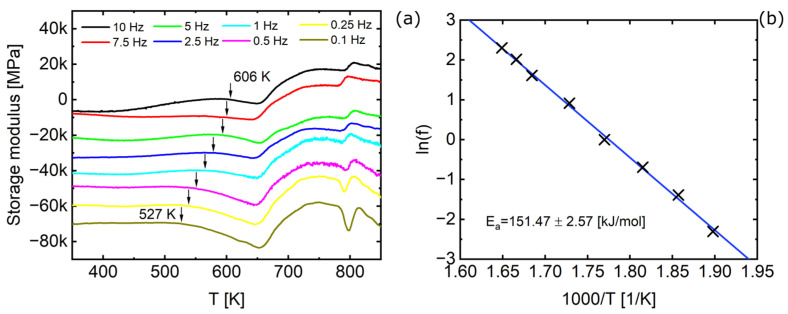
DMA measurements for Fe_81.2_Co_4_Si_0.5_B_9.5_P_4_Cu_0.8_ before HPT deformation (**a**) Storage modulus measured by DMA with varying frequency (0.1–10 Hz) at a heating rate of 10 K/min (only selected numeric values given due to visibility) (**b**) Arrhenius plot of the glass transitions from Figure 8a for the determination of the activation enthalpy of the glass transition.

**Figure 9 materials-16-01260-f009:**
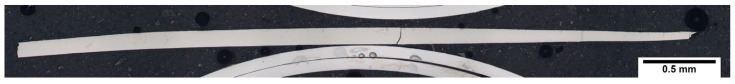
Cross-section of alloy B (Fe_81.2_Co_4_Si_0.5_B_9.5_P_4_Cu_0.8_) after HPT deformation at room temperature to *N* = 10.

**Figure 10 materials-16-01260-f010:**
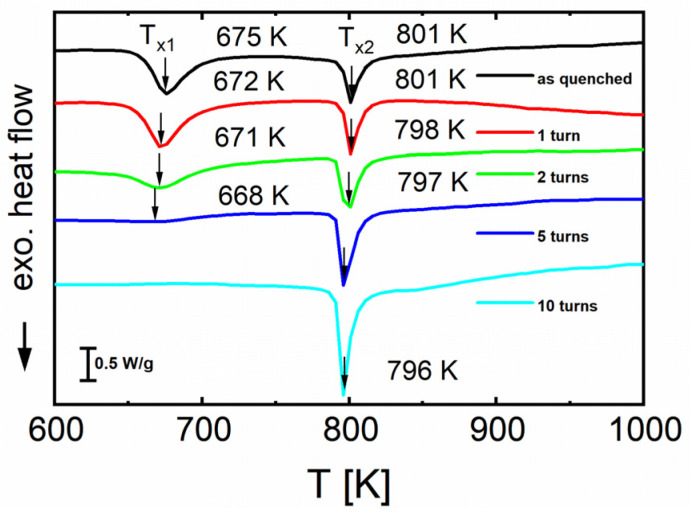
Constant-rate heating DSC scans (heating rate 20 K/min) of alloy B (Fe_81.2_Co_4_Si_0.5_B_9.5_P_4_Cu_0.8_) before (as-quenched) and after HPT deformation at *N* = 1, 2, 5, and 10.

**Figure 11 materials-16-01260-f011:**
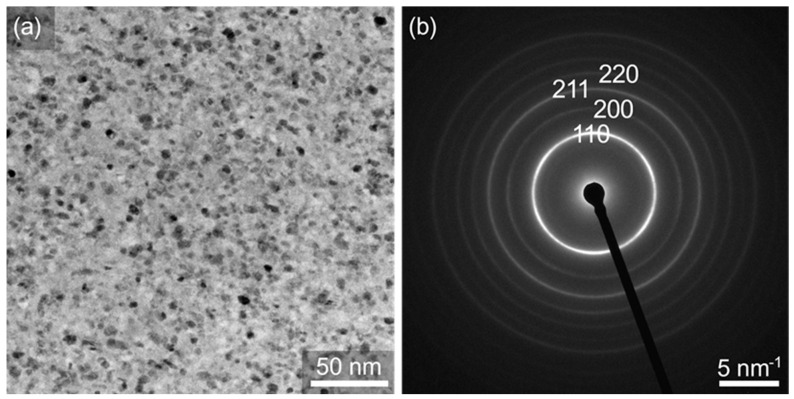
Bright-field STEM micrograph (**a**) and selected-area diffraction pattern (**b**) of alloy B (Fe_81.2_Co_4_Si_0.5_B_9.5_P_4_Cu_0.8_) after deformation at room temperature, at a pressure of 3 GPa, for 10 turns at a rotation speed of 0.1 turns/min.

**Figure 12 materials-16-01260-f012:**
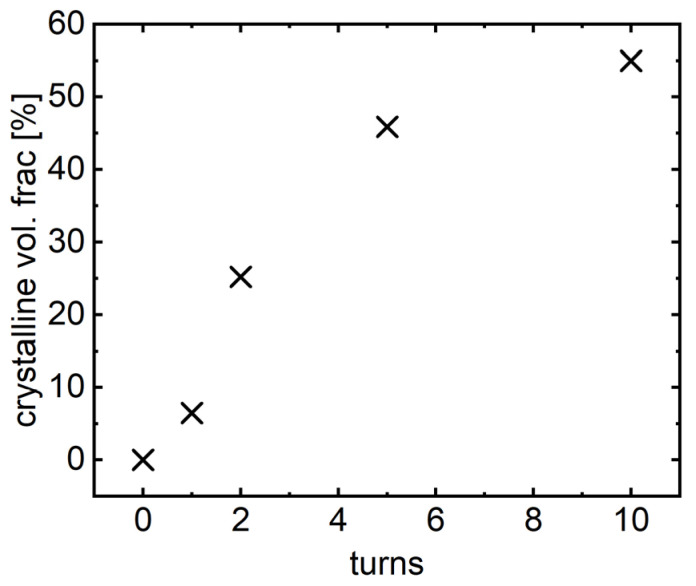
Evolution of the crystalline volume fraction in alloy B (Fe_81.2_Co_4_Si_0.5_B_9.5_P_4_Cu_0.8_) vs. the number of HPT turns calculated from the DSC data (for details, see text).

**Figure 13 materials-16-01260-f013:**
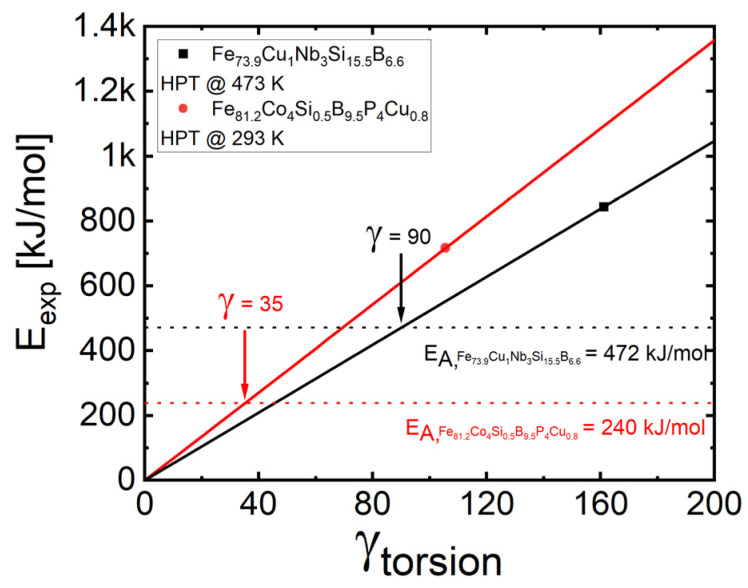
Expended energy *E_exp_* for alloys A (black) and B (red), during HPT, plotted vs. the HPT strain *γ_torsion_* with the critical levels of respective activation enthalpies *E_A_*, for the first stage of crystallization indicated. For alloy B, crossing occurs at a significantly smaller HPT strain than for alloy A.

**Table 1 materials-16-01260-t001:** HPT deformation parameters.

	Fe_73.9_Cu_1_Nb_3_Si_15.5_B_6.6_ (Alloy A)	Fe_81.2_Co_4_Si_0.5_B_9.5_P_4_Cu_0.8_ (Alloy B)
Number of stacked ribbons	50	30
Deformation temperature	200 °C (473 K)	Room temperature (293 K)
Deformation speed	0.2 turns/min	0.1 turns/min
Deformation pressure	4 GPa	3 GPa
Number of turns	1, 2, 5 and 10	
Final thickness range	70–156 µm	100–238 µm

## Data Availability

The raw data presented in this study will be made available on reasonable request to the corresponding author.

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
