# Peer review of "Can Severe Plastic Deformation Tune Nanocrystallization in Fe-Based Metallic Glasses?"

_materials, 2023, doi:10.3390/ma16031260_

Round 1

Reviewer 1 Report

This is an interesting article on the application of severe plastic deformation to two amorphous alloys. The manuscript can be accepted for publication after addressing the following issues.

1. Introduction

- For readers that are not in the field of severe plastic deformation, please cite appropriate reading references. Perhaps, a recent review paper in Mater Res Lett written by 50 experts in the field can be a good option.

- I also suggest citing an appropriate reference for the high-pressure torsion method.

2. Experimental procedure

- Please mention how instrumental broadening of XRD was considered in equation (2). Was a standard sample used?

- Moreover, since the hardness of materials is quite high and the applied pressures are rather low, please also comment if any slippage between the sample and anvils occurred.

- I also suggest mentioning how many ribbons were stuck for each HPT process.

- In Table 1, thickness ranges should be written as 70-156 and 100-238 microns.

- The position for sampling by FIB should be mentioned by considering that the microstructure may not be uniform from the disc center to the edge.

3. Results

- On page 5, it was mentioned that the shear strain after 10 HPT rotations is 3400. Was this an average strain from the center to the edge or the maximum value?

- On page 7, add a reference for Kissinger method. In the caption of Fig. 7, Tp and beta should be also defined.

- On page 9, how did you determine that 50% of the sample is crystalline? This should be a hard task using TEM. Please add some comments to the text.

4. Discussion

- The authors developed some discussion on the different behavior of the two alloys: strain-induced crystallization in one alloy but amorphization in the other one. The suggested reasons for crystallization seem reasonable, but I suggest gathering some data from the literature on the application of HPT to different amorphous alloys and discussing the occurrence or nonoccurrence of crystallization in these alloys by considering the suggested reasons in the current study. Since this issue has been a topic of discussion for about two decades, such information preferably in the form of a table can be quite interesting.

5. Conclusions

- The section was written well, but “because of the larger number” is ambiguous. Please check the sentence.

Author Response

Please refer to the attached response letter!

Reviewer 2 Report

This paper reports a novel approach for inducing nanocrystallization in soft magnetic metallic glass ribbons. While the report is interesting and deserves to be published, there are some issues that should be addressed before publication:

1- the term "SPD deformation" in the title is not correct because SPD itself includes the deformation term. So, the term "deformation" coming after SPD is superfluous and should be removed.

2- In Table 1, the number of stacked ribbons in alloy A is 50. Considering the thickness of 20 microns for each ribbon, the expected final thickness should be ~1000 microns. Why the final thickness range is quite smaller?

3- Why is HPT for alloy B done at room temperature but for alloy A the deformation temperature of 200 deg. C is used?

4- Two different activation energy units are used throughout the text, which are kJ/mol and kJ/g. Only one of these units should be used in the text.

5- Fig. 3 should be revised because the crystalline planes are shown in the figure while there is no sharp crystalline peak.

4- Fif. 6 should also be revised. It is very strange that the intensity of 10 turn sample is almost similar to the amorphous as-spun sample. I think the intensities of the XRD patterns are not correctly normalized in this figure.

5- The standard deviations for activation energy determination in Fig. 7b should be given.

Author Response

(The authors gave the same response as above.)

Reviewer 3 Report

Review report

In this manuscript, the authors discuss the effect of HPT processing on the nanocrystalline structure in Fe-based metallic glass. Although I am interested in the manuscript, I comment on the following issues to improve the manuscript.

Major comments

1. It wasn't easy to understand why the deformation temperature of Alloys A and B. Thus, the authors should mention it in more detail.

2. Is it possible to evaluate the activation enthalpy, Ea, for crystallization in alloy A? To compare Ea in alloys A and B, the readers can understand both alloys' activation processes more quantitatively.

Minor comments

The authors should check the manuscript carefully and thoroughly and revise it. Even that I realized, the following points are seen;

1. Title: no need for "deformation" because the "deformation" in the "SPD deformation" is repeated.

2. L.77: Why the order of reference no. is irregular?

3. L.99: The x In "Tx,n," isn't "c" corresponding to "crystallization" more suitable than "x"? And the authors should explain the meaning of the "n."

4. L.142: The "-2" in “30-2kV” is seemed not to need.

5. L.143: Since no STEM images are shown in the manuscript, "STEM" should be revised as "TEM."

6. L.154: "see, Fig.2," I suggest that the authors do not use a conversational style in the manuscript.

7. L.192: Is "Fig.11" correct?

8. In Ref.14, is "992" correct?

9. Figure 3; the space in indices should be narrow. And why don't you add the arrows at the ideal positions (q) of indices?

10. Figs.7 and 8: why don't you indicate T (not 1/T) as the second X-axis?

11. I am unfamiliar with the description of "Tab.X." Table X is better than "Tab.X."

12. Tab.2: I suggest that the torsional shear strain is revised to γtorsion, as the same description in Fig.13.

Author Response

(The authors gave the same response as above.)

Round 2

Reviewer 1 Report

The manuscript has been revised carefully and can be published in its current form.

Reviewer 2 Report

I think the author's have well responded to the comments and I do support publication of the manuscript.